# Celiac Disease and the Risk of Cardiovascular Diseases

**DOI:** 10.3390/ijms24129974

**Published:** 2023-06-09

**Authors:** Yichen Wang, Bing Chen, Edward J. Ciaccio, Hani Jneid, Salim S. Virani, Carl J. Lavie, Jessica Lebovits, Peter H. R. Green, Chayakrit Krittanawong

**Affiliations:** 1Mercy Internal Medicine Service, Trinity Health of New England, Springfield, MA 01104, USA; 2Department of Gastroenterology and Nutrition, Geisinger Medical Center, Danville, PA 17821, USA; 3Department of Medicine, Celiac Disease Center, Columbia University College of Physicians and Surgeons, New York, NY 10032, USA; 4Division of Cardiology, University of Texas Medical Branch, Houston, TX 77030, USA; 5Section of Cardiology and Cardiovascular Research, Department of Medicine, Baylor College of Medicine, Houston, TX 77030, USA; 6Office of the Vice Provost (Research), The Aga Khan University, Karachi 74800, Pakistan; 7John Ochsner Heart and Vascular Institute, Ochsner Clinical School, University of Queensland School of Medicine, New Orleans, LA 70121, USA; 8Cardiology Division, NYU School of Medicine, New York, NY 10016, USA

**Keywords:** celiac disease, cardiovascular disease, gluten-free diet, risk factors, myocardial infarction

## Abstract

Celiac disease (CD) is a chronic autoimmune disorder that affects the small intestine in genetically predisposed individuals. Previous studies have investigated the potential link between CD and cardiovascular disease (CVD); however, the findings have been inconsistent. We aimed to provide an updated review of the literature on the association between CD and CVD. PubMed was searched from inception to January 2023 using keywords including CD, cardiovascular disease, coronary artery disease, cardiac arrhythmia, heart failure, cardiomyopathy, and myocarditis. We summarized the results of the studies, including meta-analyses and original investigations, and presented them according to the different forms of CVD. Meta-analyses published in 2015 provided mixed results regarding the relationship between CD and CVD. However, subsequent original investigations have shed new light on this association. Recent studies indicate that individuals with CD are at a higher risk of developing overall CVD, including an increased risk of myocardial infarction and atrial fibrillation. However, the link between CD and stroke is less established. Further research is needed to determine the link between CD and other cardiac arrhythmias, such as ventricular arrhythmia. Moreover, the relationship between CD and cardiomyopathy or heart failure, as well as myopericarditis, remains ambiguous. CD patients have a lower prevalence of traditional cardiac risk factors, such as smoking, hypertension, hyperlipidemia, and obesity. Therefore, it is important to discover strategies to identify patients at risk and reduce the risk of CVD in CD populations. Lastly, it is unclear whether adherence to a gluten-free diet can diminish or increase the risk of CVD among individuals with CD, necessitating further research in this area. To fully comprehend the correlation between CD and CVD and to determine the optimal prevention strategies for CVD in individuals with CD, additional research is necessary.

## 1. Introduction

Celiac disease (CD) is a genetically determined autoimmune disorder characterized by an adverse reaction to gluten, a protein found in wheat, barley, and rye. The estimated prevalence of this disease in the general population is approximately 1% [1]. A higher incidence of CD has been observed in women, typically diagnosed during childhood and adolescence, or at ages 40–60, although it may present at any stage of life. CD is clinically noteworthy due to its diverse range of presentations, associated morbidity that can significantly impact patients’ quality of life, and increased mortality [2,3,4]. In recent years, there has been growing recognition of the potential for CD to cause extraintestinal manifestations, including its association with cardiovascular disease (CVD) [5,6,7]. The purpose of this review is to examine the latest evidence of the relationship between CD and CVD, as well as the underlying mechanisms involved.

## 2. Clinical Presentation and Diagnosis

The clinical presentation of CD varies widely, ranging from asymptomatic to symptomatic individuals. Common symptoms include abdominal pain, bloating, diarrhea, constipation, weight loss, fatigue, and anemia [8]. Individuals may also present with extraintestinal symptoms, such as skin rashes (i.e., dermatitis herpetiformis), as well as osteoporosis, infertility, and neurological symptoms [9]. Diagnosis of CD requires a combination of serological and histological evaluations [10]. Serological tests, including anti-tissue transglutaminase antibodies, are used to screen for and diagnose CD. Positive serology results are typically confirmed through a biopsy of the small intestine, which is used to evaluate the extent of villous atrophy and crypt hyperplasia. Clinical presentation, genetics, and family history are also taken into consideration during the diagnostic process [11].

## 3. Pathophysiology of Celiac Disease

CD is a chronic autoimmune enteropathy that is characterized by a unique genetic predisposition defined by the presence of HLA-DQ2 and/or HLA-DQ8 genes and the production of specific autoantibodies [12,13]. The presence of HLA-DQ2 and/or HLA-DQ8 is a necessary condition for the development of CD [14]. These genes facilitate the presentation of gluten-derived peptides to CD4+ T cells, initiating an immune response [15]. The enzyme tissue transglutaminase (TG2) deamidates certain gluten peptides, increasing their affinity to HLA-DQ2 or HLA-DQ8. This results in a more vigorous CD4+ T-helper 1 T-cell activation, which can lead to the activation of cellular and humoral immune mechanisms, leading to damage of the small intestinal villi and subsequent malabsorption of nutrients. The destruction of the villi results in decreased surface area for absorption, leading to a reduced ability to absorb essential nutrients and water. The immune response also stimulates the production of antibodies, such as anti-tissue transglutaminase antibodies, which play a crucial role in the diagnosis of CD. Gluten sensitivity and the resulting immune response may account for the systemic reactions observed in CD [12,16,17].

## 4. Association between Celiac Disease and Cardiovascular Diseases

### 4.1. CD and CVD Risk Factors

In a retrospective cross-sectional study conducted in Sweden, 1075 individuals with biopsy-confirmed CD were compared to a 1:5 matched sample from the general population [18]. The results indicated that patients with CD were less likely to be active smokers, with an OR of 0.74 (95% confidence interval (CI) 0.56 to 0.98). Additionally, individuals with CD had a lower mean body mass index (BMI) of 24.6 compared to 26.1 in the general population (*p* < 0.001). While there was a reduced likelihood of hypertension in the CD group (OR = 0.80, 95% CI 0.64 to 1.00, and *p* = 0.05), the likelihood of having diabetes was similar (OR = 1.08, 95% CI 0.82–1.43). Notably, the risk of insulin-dependent diabetes was higher in the CD group (OR = 1.50, 95% CI 1.03–2.17). The total cholesterol level at the first hospitalization with myocardial infarction was lower in patients with CD (4.7 vs. 5.1 mmol/L, *p* < 0.001).

Similarly, in the prospective analysis of CD in the UK Biobank database, CD participants were found to have a lower BMI, consumed less alcohol, were less likely to smoke, had lower total cholesterol, had lower mean systolic blood pressure, were less likely to be diagnosed with type 2 diabetes, and were more likely to be diagnosed with type 1 diabetes when adjusted for age, sex, socioeconomic status, education, and ethnicity [19]. Furthermore, the ideal cardiovascular disease risk score [20] was significantly higher in individuals with CD (23.3, 95% CI 21.6 to 25.1) compared with the general population (14.3, 95% CI 14.2 to 14.4).

Another retrospective study of 1:5 age- and sex-matched 3790 CD patients to controls found a lower rate of hypertension (11% vs. 15%, OR 0.68, and 95% CI 0.60–0.76) and hyperlipidemia (3.0% vs. 4.8%, OR 0.58, and 95% CI 0.47–0.72) [21].

Regarding atherosclerosis, a critical intermediary stage in the development of ischemic heart disease, multiple studies have demonstrated that patients with CD exhibit augmented intima-medial thickness, decreased elasticity of the ascending aorta, and impaired endothelial function, possibly due to heightened systemic inflammation [22,23,24].

Overall, CD appears to be associated with a reduced burden of CVD risk factors such as lower BMI, lower rates of smoking, lower cholesterol levels, lower mean systolic blood pressure, and a lower likelihood of type 2 diabetes. However, it should be noted that CD patients do have a higher risk of type 1 diabetes and are more likely to develop atherosclerosis, likely as a result of systemic inflammation. In addition, patients with CD frequently have low high-density lipoprotein levels that may contribute to atherosclerosis. HDL levels frequently rise after implementing a gluten-free diet [25]. This implies that conventional risk factors used to predict CVD may not be as relevant in CD patients, and more careful monitoring is necessary for this population.

### 4.2. Celiac Disease and CVD

We summarize recent studies on the relationship between CD and the risk of CVD as a compound outcome in Table 1. The relationship between CD and CVD is illustrated in Figure 1. Studies have examined the potential link between CD and CVD. While two meta-analyses published in 2015 yielded mixed results, subsequent original investigations have shed new light on the association.

Two meta-analyses published in 2015 analyzed evidence predating that year [28,29]. Heikkila et al., incorporated 13 studies [21,30,31,32,33,34,35,36,37,38,39,40,41] and yielded a pooled hazard ratio of 1.05 (95% CI 0.93–1.19) for incident ischemic heart disease, which was not statistically significant. However, the analysis noted the presence of substantial heterogeneity (I-squared 69.2%) within the studies included. Following the exclusion of two studies of dermatitis herpetiformis, the hazard ratio was found to be 1.15 (95% CI 1.04–1.28), a statistically significant finding. Emilsson et al., analyzed 10 studies [2,21,31,32,33,34,35,36,38,42] and found there was an increased risk of stroke (OR 1.11, 95% CI 1.02–1.20), but not myocardial infarction (OR 1.12, 95% CI 0.83–1.40) or cardiovascular death (OR 1.12, 95% CI 0.96–1.29) [29]. Since then, four further original investigations have been undertaken to explore the potential link between CD and CVD [4,19,43,44].

In a prospective analysis of a UK cohort from the UK Biobank database, consisting of 469,095 adults aged 40–69 years, of whom 2083 were diagnosed with CD (identified by a combination of self-report with verbal interviews and hospital inpatient diagnosis) between 2006 and 2010, an augmented incidence rate of CVD was observed in individuals with CD compared to those without the disease over a median follow-up of 12.4 years [19]. The incidence rate was noted to be 9.0 per 1000 person years (95% CI 7.9–10.3) in patients with CD and 7.4 per 1000 person years (95% CI 7.3–7.4) in individuals without CD. The unadjusted hazard ratio was calculated to be 1.27 (95% CI 1.11 to 1.45, *p* < 0.001) and persisted even after adjusting for demographic information and lifestyle factors (sex, Townsend score, education, region, year of birth, year of recruitment, ethnicity, smoking, alcohol consumption, and physical activity). Notably, further adjustment for CVD factors, including a family history of CVD, total cholesterol, glucose, antihypertensive use, cholesterol-lowering medication use, and diabetes, strengthened the association between CD and CVD (adjusted hazard ratio 1.44, 95% CI 1.26–1.65, and *p* < 0.001).

In a retrospective population-based cohort study involving 49,829 Swedish patients with histopathology-diagnosed CD between 1969 and 2017 and matched with the general population on age, sex, county, and calendar period at a ratio of 1:5, an increased risk of death from CVD was observed in individuals with CD (3.5 vs. 3.4 per 1000 person years, HR 1.08, and 95% CI 1.02–1.13) [4]. Further adjusting for education, Nordic country of birth, type 1 diabetes, autoimmune thyroid disease, rheumatoid arthritis, and inflammatory bowel disease, however, yielded no statistically significant HR (1.03, 95% CI 0.98–1.08). Another population-based retrospective study was conducted with the objective of assessing the association between 19 autoimmune diseases and the risk of CVD in a UK cohort of 2,200,937 individuals [43]. Among the cohort, 24,895 individuals with CD were identified using various diagnostic codes. The study found that individuals with CD had a higher risk of CVD compared to a cohort matched for age, calendar year, sex, socioeconomic status, region, and being free of autoimmune disease at any time, as indicated by an HR of 1.50 (95% CI 1.33–1.69).

A cross-sectional study using the National Inpatient Sample (NIS) within the United States from 2005 to 2014 was conducted to explore the prevalence of CVD among 227,172 adults with CD compared to 1:5 matched non-celiac controls [44]. The results indicate that young women under the age of 40 years exhibited a significantly higher prevalence of CVD and were at increased odds of developing this condition. This finding was not observed in other age and sex groups.

In summary, while earlier meta-analyses with high heterogeneity have yielded mixed results on the potential link between CD and CVD, more recent original investigations suggest a higher risk of CVD in individuals with CD. These results are of importance clinically as prompt identification and treatment of CVD risk factors in individuals with CD can alleviate the CVD burden in this population. It is worth noting that the majority of these studies have been conducted in Nordic and Western European countries, as well as North America. Therefore, further research conducted in other regions of the world is necessary to better understand this correlation and to establish optimal strategies for CVD prevention and management in individuals with CD.

### 4.3. CD and Myocardial Infarction

In a retrospective cross-sectional study in 2013 of a Sweden population of 1075 comparing patients with biopsy-confirmed celiac disease to a 1:5 matched general population, researchers found higher odds of myocardial infarction (2.1% vs. 1.8%, *p* < 0.001) [18]. However, it was observed that ST-elevation myocardial infarction and angiography-confirmed stenosis (OR 0.80, 95% CI 0.66–0.97), as well as three-vessel disease not affecting the left main stem (OR 0.73, 95% CI 0.57–0.94), were less common in CD patients while the rest of the angiography findings were not statistically significant. In addition, as finding the risk of myocardial infarction in CD was not the primary objective of this research, no in-depth statistical analysis, including adjusting for covariates, was conducted. The authors conducted a meta-analysis in 2015 [29], which included four original investigations [2,21,32,36]. The analysis revealed a trend toward increased risk of myocardial infarction in patients with CD; however, this trend was not statistically significant (relative risk [RR] 1.12, 95% CI 0.83–1.40). Since then, Conroy et al., found in a prospective cohort study of 2083 patients in 2023 an increased hazard ratio of 1.5 (95% CI 1.30 to 1.75) for ischemic heart disease and 1.59 (95% CI 1.25 to 2.01) for myocardial infarction in individuals with CD [19]. Therefore, while some studies have shown higher odds of myocardial infarction in patients with biopsy-confirmed CD compared to the general population, others have not found a statistically significant association. However, a recent prospective cohort study in 2023 suggests an increased hazard ratio for ischemic heart disease and myocardial infarction in individuals with CD. Further research is needed to fully understand the potential link between CD and cardiovascular disease.

### 4.4. CD and Stroke

In a meta-analysis of two studies [21,31], Emilsson et al., discovered that there was an association between CD and a heightened risk of stroke (OR, 1.11, 95% CI 1.02–1.20) [29]. This finding, however, was heavily impacted by the larger of the two studies, which was a retrospective analysis of 28,676 individuals with pathology-confirmed CD [31]. After adjusting for various risk factors, such as type 1 diabetes, rheumatoid arthritis, the use of hypertensive medication, and lipid-lowering therapy, the association was no longer statistically significant (HR 1.11, 95% CI 0.96–1.29). A recent study of a prospective cohort of 2083 patients with CD in the United Kingdom reported that CD was not significantly associated with an elevated risk of stroke after accounting for lifestyle factors (HR 1.15, 95% CI 0.86–1.53) and major risk factors for cardiovascular disease (HR 1.20, 95% CI 0.89 to 1.60) [19]. Furthermore, a considerable number of studies employed a composite outcome for stroke without distinguishing between ischemic and hemorrhagic variants, thereby introducing substantial heterogeneity. One sole investigation distinguished ischemic stroke from the collective stroke outcome and discovered a marginally significant association between CD and ischemic stroke, with a hazard ratio (HR) of 1.10 and a 95% CI ranging from 1.00 to 1.21, although the exact *p*-value was not provided [31]. Therefore, the association between CD and stroke appears to be less established, with studies showing a trend toward increased risk. Additional studies are needed to further investigate this potential association.

### 4.5. CD and Cardiomyopathy/Heart Failure

Curione et al., conducted a study on a cohort of 52 patients who were diagnosed with idiopathic dilated cardiomyopathy. Among these patients, three individuals (5.8%) were found to have biopsy-proven CD [45]. The researchers then compared this prevalence rate to that reported in the general population (1.8%) from another study [46], and concluded there was a statistically significant increase in the prevalence of CD in the cohort of patients with idiopathic dilated cardiomyopathy (*p* < 0.001). Another preliminary study in Danish patients with CD found a high incidence of cardiomyopathy in CD patients [47]. In a subsequent study, a higher rate of endomysial antibody positivity was found in 1.9% of 642 patients on the waiting list for heart transplantation in Italy compared to results from a survey conducted in the general Italian population (0.35%, RR 5.3, and 95% CI 2.8–10.3). Similarly, a study of Brazilian heart transplant recipients found a high seroprevalence of CD markers. Further evidence suggesting an association between CD and heart failure can be found in a prospective evaluation of cardiac function in 50 pediatric CD patients (mean age 4.2 ± 1.1 years) [48]. This study demonstrated that the left ventricular end-diastolic diameter was larger (35.33 ± 0.87 mm vs. 32.90 ± 0.91 mm, *p* = 0.04) and the risk of reduced left ventricular ejection fraction (<55%) was higher (20% vs. 0%, *p* = 0.01) in untreated CD patients at the time of enrollment (reflecting a cross-sectional approach) compared to healthy controls.

In contrast, a study using the Swedish national registry of 9363 children and 4969 adults with CD found no association between childhood-diagnosed cardiomyopathy (HR 0.8, 95% CI 0.2–3.7) or adult-diagnosed CD (HR 1.7, 95% CI 0.4–6.5) [49]. In a retrospective cross-sectional study of 1075 Swedish patients with biopsy-confirmed CD compared to a 1:5 matched general population, the left ventricular ejection fraction (LVEF) was found to be significantly better in CD patients (LVEF > 49%: 60.1% vs. 50.5%, *p* = 0.049) [18].

In general, the connection between CD and cardiomyopathy or heart failure appeared to be more frequently reported in case–control studies based on heart transplant registries rather than in cohort studies that identified cardiomyopathy or heart failure in patients with CD. This difference could be partially attributed to the overall low occurrence of cardiomyopathy, which provides case–control studies with a statistical edge. However, it is also possible that bias, such as the underdiagnosis of CD in the general population, may play a role. Consequently, the conclusion regarding this matter remains ambiguous.

### 4.6. CD and Cardiac Arrhythmias

Autoimmune diseases, including celiac disease (CD), are frequently associated with cardiac arrhythmias [50]. In a retrospective study by West et al., which included 3790 patients with CD and an age- and sex-matched control group in a 1:5 ratio, a slightly elevated but not statistically significant risk of atrial fibrillation was observed in patients with CD (2.1% vs.1.7%, OR 1.26, and 95% CI 0.97–1.64) [21]. In another study of 28,637 patients diagnosed with CD through biopsy, matched by age and sex to the general population, an increased risk of atrial fibrillation was reported (HR 1.34, 95% CI 1.24–1.44), and this association persisted after adjusting for various covariates, including type 1 diabetes, autoimmune thyroid disease, rheumatoid arthritis, education, and country of birth (Nordic vs. non-Nordic) [51]. In this study focused on autoimmune myocarditis, four out of thirteen patients with comorbid CD and autoimmune myocarditis exhibited ventricular arrhythmia [52]. Notably, the arrhythmia improved with the implementation of a gluten-free diet alone. Additionally, 17 out of 53 untreated CD patients had a prolonged QTc (>440 milliseconds), which may explain the higher risk of ventricular arrhythmia [53]. The association between CD and cardiac conduction disturbances, however, is understudied, with only case reports available [54,55,56,57]. Overall, while abundant evidence suggests an association between CD and atrial fibrillation, further exploration is required to determine the association between CD and other cardiac arrhythmias, such as ventricular arrhythmia and conduction disturbances.

### 4.7. Celiac Disease and Myopericarditis

Frustaci et al., conducted a case–control study and observed a significantly elevated prevalence of biopsy-confirmed CD among the 187 patients with clinically and histologically diagnosed myocarditis, compared to the 306 individuals in the control population (4.4% vs. 0.6%, *p* < 0.003) [52]. In contrast, Elfstrom et al., investigated the association between CD and myocarditis or pericarditis in a Swedish national registry of 9363 children and 4969 adults with CD [49]. They found no significant association between CD and myocarditis in childhood (HR 0.2, 95% CI 0.0–1.5) or adulthood (HR 2.1, 95% CI 0.4–11.7), or between CD and pericarditis in childhood (HR 0.4, 95% CI 0.1–1.9) or adulthood (HR 1.5, 95% CI 0.5–4.0). These contrasting results may be due to the rarity of myopericarditis in both populations, and further studies are needed to investigate this topic.

## 5. Mechanism and Prevention of CVD

One hypothesis posits that the link between CD and CVD is attributable to heightened systemic inflammation. This notion is buttressed by increased CVD risk in a number of autoimmune disorders [43], as well as by plausible mechanisms that link systemic inflammation to elevated atherosclerosis seen in both CD and other autoimmune conditions [58]. Given the theoretical association between systemic inflammation and CVD risk, speculation has arisen that a gluten-free diet could reduce inflammation and thereby mitigate CVD risk. However, this was not borne out in CD patients in a large study that looked at the risk of coronary heart disease according to the amount of gluten consumed. Those in the lowest gluten consumption group had a higher rate of heart disease [59]. A gluten-free diet is not universally recognized as a healthy option and may actually contribute to an increased risk of heart disease. This risk may be attributed to the increased consumption of saturated fat and sugar, as well as the decreased intake of complex whole grains associated with a gluten-free diet [59,60]. In a systematic review of 27 studies published in 2018, investigators sought to determine whether gluten-free diets could reduce CVD risk in patients with CD [61]. However, no such link was found. It is important to note that the overall evidence on this topic remains poor, as only one study utilized a control group. Consequently, further research is required to investigate dietary interventions aimed at reducing CVD risk in individuals with CD. The gluten-free diet is trendy, and potential health advantages to the general population have not been confirmed in careful studies [62,63,64].

## 6. Molecular Mechanisms Linking CD and CVD

Inflammation plays a central role in the pathogenesis of both CD and CVDs. The ingestion of gluten in CD patients leads to the activation of the immune system, resulting in the production of proinflammatory cytokines such as tumor necrosis factor-alpha (TNF-α), interleukin-1 (IL-1), and IL-6 [65]. These cytokines contribute to the development of intestinal inflammation, villous atrophy, and crypt hyperplasia observed in CD [65]. Moreover, systemic inflammation is a hallmark of atherosclerosis, a major contributor to CVDs. The aforementioned proinflammatory cytokines are involved in the initiation and progression of atherosclerotic plaques [66]. Therefore, it is plausible that the systemic inflammation observed in CD patients could contribute to an increased risk of CVDs.

Endothelial dysfunction, characterized by impaired vasodilation and proinflammatory and prothrombotic endothelial cell phenotypes, is an early marker of atherosclerosis [67]. Studies have shown that CD patients exhibit endothelial dysfunction, as evidenced by impaired flow-mediated dilation and increased levels of endothelial activation markers, such as von Willebrand factor and soluble intercellular adhesion molecule-1 (sICAM-1) [68]. These findings suggest that the endothelial dysfunction observed in CD patients may contribute to the development of CVDs. Additionally, shared genetic susceptibility factors, such as human leukocyte antigen (HLA) haplotypes, could further link CD and CVDs [69].

In conclusion, the molecular mechanisms underlying the association between CD and CVDs involve shared inflammatory pathways, endothelial dysfunction, and genetic susceptibility factors (Figure 2). A better understanding of these mechanisms could lead to the development of targeted therapies and preventive strategies for both CD and CVDs. Further research is needed to elucidate the complex interplay between these factors and to establish the causal relationship between CD and CVDs. Longitudinal and interventional studies, as well as the assessment of biomarkers and genetic factors, will be crucial in advancing our knowledge in this field. Additionally, investigating the potential impact of a gluten-free diet on CVD risk factors and outcomes in CD patients will provide valuable insights into the role of dietary modifications in mitigating the risk of CVDs. Collectively, these efforts will contribute to a comprehensive understanding of the relationship between CD and CVDs, ultimately leading to improved clinical management and prevention strategies for these prevalent and debilitating diseases.

## 7. Conclusions

In conclusion, recent original investigations suggest that individuals with CD may have a higher risk of CVD compared with the general population. While earlier meta-analyses yielded mixed results, several studies have found an increased risk of ischemic heart disease, myocardial infarction, and atrial fibrillation in patients with CD. It is worth noting that CD patients may have a lower prevalence of traditional CVD risk factors such as hypertension and smoking, but a higher incidence of insulin-dependent diabetes. Prompt identification and treatment of CVD risk factors in individuals with CD can alleviate the CVD burden in this population. However, traditional risk reduction measures may not be as effective in this population, considering the unique risk factor profile in this group. Moreover, the impact of a gluten-free diet on CVD risk in CD patients is not yet established, and further investigation is necessary, as it may even theoretically increase the risk of CVD due to its ingredient components. Given the high morbidity and mortality associated with CVD, the potential link between CD and CVD requires continued investigation. Although a majority of studies have been conducted in Western European countries and North America, further research is necessary to establish optimal strategies for CVD prevention and management in individuals with CD in other regions of the world. Therefore, it is crucial to continue investigating this correlation to improve our understanding and management of this important health issue.

## Figures and Tables

**Figure 1 ijms-24-09974-f001:**
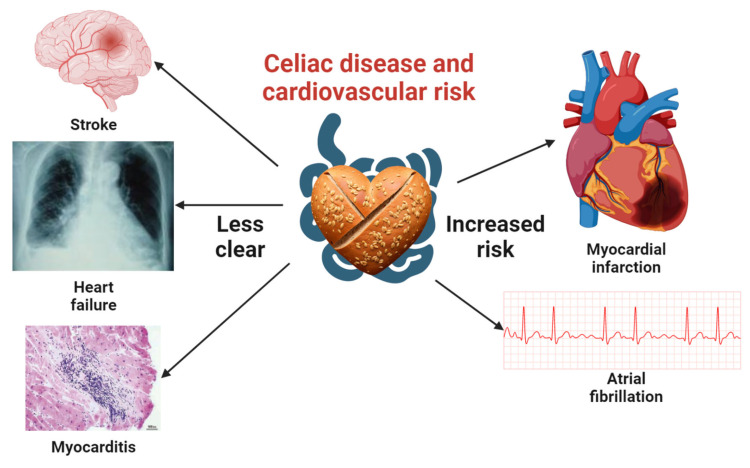
Comprehensive visualization of the association between celiac disease and cardiovascular disease risk. This central illustration highlights the well-documented links between celiac disease, myocardial infarction, and atrial fibrillation. It also emphasizes the need for additional studies to clarify the connections with stroke, heart failure, and myocarditis. Icons representing myocardial infarction, atrial fibrillation, and stroke were obtained from Biorender.com, while those for heart failure and myocarditis were sourced from open-access articles on PubMed Central, identified through an Open-i search [26,27]. A heart-shaped barley bread icon, created using the Stable-Diffusion text-to-image model, signifies the context of celiac disease in this investigation.

**Figure 2 ijms-24-09974-f002:**
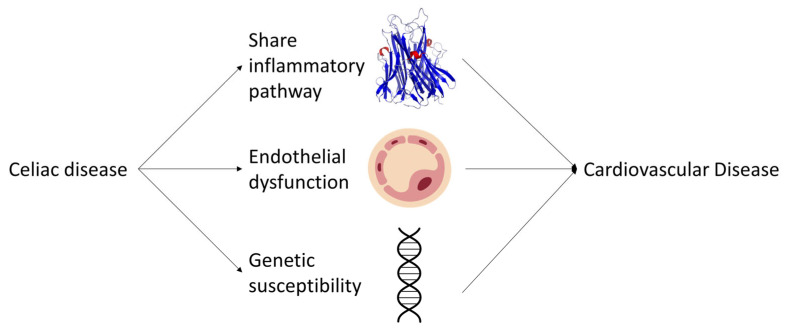
Schematic representation of the potential pathways linking celiac disease to an increased risk of cardiovascular diseases.

**Table 1 ijms-24-09974-t001:** Overview of key studies evaluating cardiovascular disease risks in celiac populations from 2000 to present.

First Author	Publication Year	Type of Study	Method Used to Identify Celiac Disease	Study Duration	Patients, n	Country	Cardiovascular Disease-Related Outcomes	Effect Size	Adjusted Variables
Conroy	2023	Prospective cohort	Combination of self-report with verbal interview and hospital inpatient diagnosis	Patient enrolled from 2006 to 2010, with a median follow-up of 12.4 years	2083 celiac disease, 467,012 without celiac disease	UK	Composite cardiovascular disease incidents (ischemic heart disease, myocardial infarction, and stroke) identified by diagnostic codes from hospital inpatient and death records. Primary outcome of this study.	Adjusted HR: 1.44 (1.26–1.65)	Sex, Townsend score, education, region, year of birth, year of recruitment, ethnicity, smoking, alcohol consumption, physical activity, family history of CVD, total cholesterol, glucose, antihypertensive use, cholesterol-lowering medication use, and diabetes
Conrad	2022	Retrospective cohort	Various diagnostic codes	Celiac disease was diagnosed between 2020 and 2017. Median follow-up 6.2 years (IQR 2.7–10.8)	24,895 celiac disease, 1:5 matched to general population	UK	Composite cardiovascular disease incidents (aortic aneurysm, atrial fibrillation and flutter, supraventricular arrhythmias, conduction system disease, myocarditis and pericarditis of non-infectious origin, peripheral arterial disease, infective endocarditis, stroke (ischemic and hemorrhagic) or TIA, valve disorders (excluding congenital and rheumatic), and venous thromboembolism. Celiac disease belongs to a subgroup analysis of this study on 19 autoimmune diseases.	HR: 1.50 (1.33–1.69)	Matched by age, sex, socioeconomic status, region, and calendar year. No further adjustment of comorbidities
Naaraayan	2021	Cross-sectional	ICD-9 diagnostic codes	Hospitalization with diagnosis of celiac disease from 2005 to 2014. No follow-up (cross-sectional)	227,172 adult hospitalizations with celiac disease, 1:5 matched to hospitalizations without celiac disease	US	ICD-9 identified atherosclerotic cardiovascular disease presence in hospital inpatient records.	Adjusted OR on matched groups: 0.98 (0.95–1.01)	Matched by age, sex, race, and calendar year to hospitalizations. Further adjusted to hypertension, diabetes, hyperlipidemia, obesity, chronic kidney disease, and smoking status
Lebwohl	2020	Retrospective cohort	Histopathologically diagnosed	Patients were diagnosed from 1969 to 2017, with a median follow-up of 12.5 years	49,829 celiac disease, 1:5 matched to general population	Sweden	ICD-identified cardiovascular disease caused death. Secondary outcome of this study.	Matched and unadjusted HR:1.08 (1.02–1.13) Matched and adjusted HR: 1.03 (0.98–1.08)	Groups matched by age, sex, county, and calendar period. Further adjustment including education, Nordic country of birth, and medical comorbidities (type 1 diabetes, autoimmune thyroid disease, rheumatoid arthritis, and inflammatory bowel disease)
Ludvigsson	2012	Cohort, possibly mixture of retrospective and prospective	Serology-positive	Patients were diagnosed from 1969 to 2008. follow-up median and range 9 (0–40) years, mean and SD 10.4 (6.4)	46,330 patients with positive celiac disease serology who underwent biopsy (regardless of biopsy results), up to 1:5 matched on age and sex to general population	Sweden	ICD-identified stroke (ischemic and hemorrhagic incidents. Primary outcome of this study.	Matched and unadjusted HR: 1.10 (1.01–1.19) Matched and adjusted HR: 1.11 (0.96–1.29)	Groups matched by age and sex. Further adjustments included type 1 diabetes, rheumatoid arthritis, use of hypertensive medication, and lipid-lowering therapy
Ludvigsson	2011	Cohort, possibly a mixture of retrospective and prospective	Small intestinal histopathology	Patients were diagnosed from 1969 to 2008. follow-up median and range 8 (0–39) years	28,190 patients with biopsy-proven CD, up to 1:5, matched on age, sex, county, and calendar year to general population	Sweden	ICD-identified composite incident ischemic heart disease (myocardial infarction and angina pectoris) from hospital inpatient and death records. Primary outcome of this study.	Matched HR 1.19 (1.11–1.28)	Matched by age, sex, county, and calendar year
Grainge	2011	Prospective cohort	Biopsy-proven	Followed-up from 1978 until death or 2006. Median and IQR 6.2 (3.2–12.9) years	1092 CD patients with at least 2 years of follow-up	Derby, UK	Cardiovascular disease caused death based on ICD-10 codes (I00-I99) assigned on death certificates. Secondary outcome of this study.	SMR 1.12 (0.82–1.50)	SMR generated using rates of cohorts stratified by 5-year age category, sex, and calendar year
Wei	2008	Community-based cohort, likely retrospective	Serology-positive or biopsy-proven	Average follow-up of 3.7 years	367 CD patients and 5537 antiendomysial antibody negative comparators	Tayside, Scotland, UK	ICD-identified composite incident cardiovascular disease (ischemic heart disease, heart failure, cerebrovascular disease, or cardiovascular death) from hospital inpatient and death records. Primary outcome of this study.	Unadjusted HR: 1.10 (0.62–1.92) Adjusted HR for gluten-free prescriptions: 2.4 (1.32–4.42)	Multiple univariate adjustments (not listed in this table), including age, sex, deprivation, diabetes, previous cardiovascular hospitalization, cardiovascular drugs, allopurinol, glucocorticoids, NSAIDs, and folic acid, all of which have statistically insignificant results
Ludvigsson	2007	Prospective cohort	ICD diagnostic codes	Unclear, patients with <1 year of follow-up were excluded	13,358 CD patients and 64,118 age- and sex-matched controls	Sweden	Hospital-based ICD-defined incident vascular diseases, including myocardial infarction, angina pectoris, heart failure, brain hemorrhage, and ischemic stroke. Primary outcome of this study.	Unadjusted HR and adjusted HR ranges from 1.24 to 1.46 for different cardiovascular diseases, all statistically significant	Matched by age and sex. Further adjustment for diabetes
Solaymani Dodaran	2007	Population-based cohort, likely retrospective	Biopsy-proven	Mean 26, median 28	285 children and 340 adults were diagnosed with CD. Mortality compared to regional expected populational mortality rates	Lothian, Scotland, UK	Cerebrovascular disease and ischemic heart disease caused deaths. ICD-defined cause of death from death registry. Secondary outcome of this study.	SMR 1.38 (0.84–2.13) and 1.20 (0.84–1.67) for cerebrovascular disease and ischemic heart disease in adults	None
West	2004	Population-based retrospective cohort	Recorded diagnosis	Unclear, 0–10 years based on Kaplan–Meier curve	3790 CD patients and 17,925 age- and sex-matched controls	UK	Incident myocardial infarction and stroke identified using recorded diagnosis.	Unadjusted HR for stroke 1.29 (0.98–1.70), for myocardial infarction 0.85 (0.63–1.13)	BMI, hypertension, and smoking were adjusted but not systemically reported
Corrao	2001	Prospective cohort	Biopsy-proven	Patients were diagnosed from 1962 to 1994. Mean follow-up 6.0 years (SD 4.9)	1072 adult patients in patient cohort: 862 patients’ parents; 862 patients’ siblings	Italy	ICD-9 identified circulatory system diseases caused deaths. Secondary outcome of this study.	SMR in celiac disease cohort 0.7 (0.3–1.5)	Standardized for age, sex, and calendar year

Abbreviations: CVD, cardiovascular disease; HR, hazard ratio; IQR, interquartile range; ICD, International Classification of Diseases; TIA, transient ischemic attack; OR, odds ratio; SD, standard deviation; SMR, standardized mortality ratio; BMI, body mass index; and NSAIDs, non-steroidal anti-inflammatory drugs.

## Data Availability

Not applicable. This is a narrative review and no new data were created or analyzed in this study.

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
