# Peer review of "Celiac Disease and the Risk of Cardiovascular Diseases"

_ijms, 2023, doi:10.3390/ijms24129974_

Round 1

Reviewer 1 Report

Dear editors:  

 It is a great honor and pleasure for me to be invited as the reviewer for this great work entitled “Celiac Disease and the Risk of Cardiovascular Diseases: An Updated Review”. Yichen Wang et al. comprehensively reviewed the association between  Celiac disease (CD) and risk of cardiovascular disease (CVD). This study topic is interesting and important, attributing to Dr. Chayakrit Krittanawong’s long-term efforts and contributions in the scientific field of CVD. Although the article is well-written, I have a number of comments concerning this study:

1.     Line 5: “Chayakrit Krittanawong and MD 8 should be rephrased.

2.     Line 29: The sentencesRecent studies indicate that individuals with CD are at a higher risk of developing overall CVD, including an increased risk of myocardial infarction and atrial fibrillation. However, the link between CD and stroke is less established. Further research is needed to determine the link between CD and other cardiac arrhythmias, such as ventricular arrhythmia” and “Recent studies have demonstrated an increased risk of overall CVD, myocardial infarction, and atrial fibrillation in individuals with CD. However, the association between CD and other CVD outcomes such as stroke, other cardiac arrhythmias, cardiomyopathy, heart failure, and myopericarditis remains uncertain (Line 31)” seem redundant in Abstract and should be rephrased.

3.     Line 55: “Clinical Presentation and Diagnosis of Celiac Disease”

4.     Line 67: Pathophysiology of Celiac Disease

5.     Abbreviation of CD should be used as follows: Page 2 Line 60, 62,70, 119 and Page 10 Line 46, 113, 213……and you can notice that line numbers are wrong.

6.     The heterogeneity is very high. Authors should divide strokes into brain hemorrhage and ischemic stroke to compare their HR ranges individually.

  Thank you for giving me the opportunity to review this interesting article. Nonetheless, Extensive editing of English language and style required may be required.

Minor editing of English language is required.

Author Response

                                                                                               NYU School of Medicine          

                                                                                               550 First Avenue, New York,

                                                                                               Chayakrit Krittanawong, MD

June 1, 2023 

Editor

Dear Editor,

We are pleased to submit the manuscript entitled, “Celiac Disease and the Risk of Cardiovascular Diseases: An Updated Review”, IJMS-2427196. We sincerely appreciate the constructive feedback provided by the reviewers and are grateful for the opportunity to revise and improve our manuscript.

In response to the comments from Reviewer 1, we have revised the manuscript accordingly. We corrected the stated errors in line 5 and line 29 and restructured the sentences to avoid redundancy. We've ensured the consistent use of abbreviations, particularly “CD”, and corrected the incorrect line numbers as pointed out. Additionally, based on their suggestion, we have divided strokes into brain hemorrhage and ischemic stroke to individually compare their HR ranges, adding depth to the discussion section.

Reviewer 2 has also offered valuable comments that have resulted in significant improvements. We have standardized the font in the abstract and added detailed discussion on CD pathophysiology focusing on HLA-DQ2/DQ8. The spacing issue pointed out in lines 95-96 has been rectified and the consistent use of abbreviations “CD” and “CI” has been ensured throughout the manuscript. To address the request for a more detailed explanation of the molecular mechanisms linking CD and CVD, we have added a comprehensive section discussing this.

Reviewer 3 pointed out an extraneous blank page in our manuscript. We have removed this page and thank the reviewer for their meticulous attention to detail.

We are grateful for the reviewers’ constructive critiques, and we believe that addressing their comments has significantly improved our manuscript. We sincerely hope that the revisions meet the reviewers’ approval and that our manuscript is found suitable for publication.

The manuscript, as submitted or its essence in another version, is not under consideration for publication elsewhere and will not be submitted elsewhere while under consideration by the International Journal of Molecular Sciences. All authors meet the criteria for authorship and that the authors will sign a statement attesting authorship, disclosing all potential conflicts of interest, and releasing the copyright should the manuscript be accepted for publication.

We are most appreciative of your consideration for publication in the and look forward to your comments.

Sincerely yours,                                                                   

Chayakrit Krittanawong, MD

Reviewer 1:

  1. Line 5: “Chayakrit Krittanawong and MD 8” should be rephrased.

We sincerely appreciate the reviewer’s attention to detail. The expression has been restructured as suggested.

  1. Line 29: The sentences “Recent studies indicate that individuals with CD are at a higher risk of developing overall CVD, including an increased risk of myocardial infarction and atrial fibrillation. However, the link between CD and stroke is less established. Further research is needed to determine the link between CD and other cardiac arrhythmias, such as ventricular arrhythmia” and “Recent studies have demonstrated an increased risk of overall CVD, myocardial infarction, and atrial fibrillation in individuals with CD. However, the association between CD and other CVD outcomes such as stroke, other cardiac arrhythmias, cardiomyopathy, heart failure, and myopericarditis remains uncertain (Line 31)” seem redundant in Abstract and should be rephrased.

We are indebted to the reviewer’s careful reading and insightful commentary. We acknowledge the redundancy and have edited the sentences accordingly to ensure brevity and clarity.  We removed the second statement. The reviewer’s constructive feedback has indeed refined the presentation of our study.

  1. Line 55: “Clinical Presentation and Diagnosis of Celiac Disease”
  2. Line 67: Pathophysiology of Celiac Disease

We sincerely thank the reviewer for the valuable suggestion. The recommended changes have been incorporated.

  1. Abbreviation of CD should be used as follows: Page 2 Line 60, 62,70, 119 and Page 10 Line 46, 113, 213……and you can notice that line numbers are wrong.

The reviewer’s keen observation is commendable. We have now consistently employed the abbreviation "CD" throughout the manuscript, except in the first instance and subtitles, aligning with conventions. Also, the line numbers have been corrected as you've kindly pointed out. Your feedback has significantly improved the clarity and readability of the text.

  1. The heterogeneity is very high. Authors should divide strokes into brain hemorrhage and ischemic stroke to compare their HR ranges individually.

We are sincerely thankful for this insightful suggestion. As the reviewer astutely observed, only one of the three referenced studies on the association between celiac disease and stroke made a distinction between ischemic and hemorrhagic strokes. We have now added a more detailed statement on page 11 line 77 to address this issue:

“Furthermore, a considerable number of studies employed a composite outcome for stroke without distinguishing between ischemic and hemorrhagic variants, thereby introducing substantial heterogeneity. Solely one investigation distinguished ischemic stroke from the collective stroke outcome and discovered a marginally significant association between CD and ischemic stroke, with a hazard ratio (HR) of 1.10 and a 95% confidence interval (CI) ranging from 1.00 to 1.21, although the exact P-value was not provided.”

The reviewer’s perceptive advice has certainly strengthened the discussion section of our study. We appreciate the dedication in fostering a rigorous and scholarly review process.

Reviewer 2:

  1. Different fonts have been used in abstract.

We appreciate your keen eye and attention to consistency. We have now standardized the font throughout the abstract, enhancing the overall presentation of our study.

  1. Line 68-69: CD pathophysiology is not discussed in detail. Since its a review focusing on CD, emphasis should be given on HLA-DQ2/ DQ8.

The reviewer’s astute observation and expert advice are greatly appreciated. We agree that more detail about the pathophysiology of celiac disease, particularly the role of HLA-DQ2/DQ8, would enhance our review. We have now elaborated on this aspect as follows:

“The presence of HLA-DQ2 and/or HLA-DQ8 is a necessary condition for the development of CD.14 These genes facilitate the presentation of gluten-derived peptides to CD4+ T cells, initiating an immune response.15 The enzyme tissue transglutaminase (TG2) deamidates certain gluten peptides, increasing their affinity to HLA-DQ2 or HLA-DQ8. This results in a more vigorous CD4+ T-helper 1 T-cell activation, which can lead to the activation of cellular and humoral immune mechanisms, leading to damage of the small intestinal villi and subsequent malabsorption of nutrients.

  1. Sollid LM, Qiao SW, Anderson RP, Gianfrani C, Koning F. Nomenclature and listing of celiac disease relevant gluten T-cell epitopes restricted by HLA-DQ molecules. Immunogenetics. 2012;64(6):455-460. doi:10.1007/s00251-012-0599-z
  2. Jones EY, Fugger L, Strominger JL, Siebold C. MHC class II proteins and disease: a structural perspective. Nat Rev Immunol. 2006;6(4):271-282. doi:10.1038/nri1805

The reviewer’s guidance has undeniably added depth to our understanding and exposition of the pathophysiology of CD in this review. We appreciate the invaluable contributions.”

  1. Line 95-96: space issue needs to rectified

The reviewer's keen eye for detail is genuinely appreciated. The spacing issue within these lines has been meticulously corrected. We believe the high standard has improved the quality of our manuscript.

  1. In many places 'Celiac disease' is used instead of CD and vice versa. Please keep it consistent (with CD). Same applies to CI/ confidence interval

We are grateful for the reviewer's discerning suggestion. The abbreviations 'CD' and 'CI' have now been uniformly applied throughout the manuscript, enhancing its readability. The reviewer's expert guidance has been instrumental in refining the presentation of our work.

  1. Molecular mechanism linking CD and CVD need to be explained in detail.

We are sincerely thankful for the reviewer's astute recommendation. The manuscript now includes a comprehensive explanation of the molecular mechanisms connecting CD and CVD:

“Inflammation plays a central role in the pathogenesis of both CD and CVDs. The in-gestion of gluten in CD patients leads to the activation of the immune system, resulting in the production of proinflammatory cytokines such as tumor necrosis factor-alpha (TNF-α), interleukin-1 (IL-1), and IL-6.63 These cytokines contribute to the development of intestinal inflammation, villous atrophy, and crypt hyperplasia observed in CD.63 Moreo-ver, systemic inflammation is a hallmark of atherosclerosis, a major contributor to CVDs. The aforementioned proinflammatory cytokines are involved in the initiation and pro-gression of atherosclerotic plaques.64 Therefore, it is plausible that the systemic inflamma-tion observed in CD patients could contribute to an increased risk of CVDs.

Endothelial dysfunction, characterized by impaired vasodilation and proinflamma-tory and prothrombotic endothelial cell phenotypes, is an early marker of atherosclero-sis.65 Studies have shown that CD patients exhibit endothelial dysfunction, as evidenced by impaired flow-mediated dilation and increased levels of endothelial activation mark-ers, such as von Willebrand factor and soluble intercellular adhesion molecule-1 (sICAM-1).66 These findings suggest that the endothelial dysfunction observed in CD pa-tients may contribute to the development of CVDs. Additionally, shared genetic suscepti-bility factors, such as human leukocyte antigen (HLA) haplotypes, could further link CD and CVDs.67

In conclusion, the molecular mechanisms underlying the association between CD and CVDs involve shared inflammatory pathways, endothelial dysfunction, and genetic susceptibility factors (Figure 2). A better understanding of these mechanisms could lead to the development of targeted therapies and preventive strategies for both CD and CVDs. Further research is needed to elucidate the complex interplay between these factors and to establish the causal relationship between CD and CVDs. Longitudinal and interventional studies, as well as the assessment of biomarkers and genetic factors, will be crucial in ad-vancing our knowledge in this field. Additionally, investigating the potential impact of a gluten-free diet on CVD risk factors and outcomes in CD patients will provide valuable in-sights into the role of dietary modifications in mitigating the risk of CVDs. Collectively, these efforts will contribute to a comprehensive understanding of the relationship between CD and CVDs, ultimately leading to improved clinical management and prevention strategies for these prevalent and debilitating diseases.

  1. Sapone A, Lammers KM, Casolaro V, et al. Divergence of gut permeability and mucosal immune gene expression in two gluten-associated conditions: celiac disease and gluten sensitivity. BMC Med. 2011;9:23. doi:10.1186/1741-7015-9-23
  2. Libby P. Inflammation in atherosclerosis. Arterioscler Thromb Vasc Biol. 2012;32(9):2045-2051. doi:10.1161/ATVBAHA.108.179705
  3. Bonetti PO, Lerman LO, Lerman A. Endothelial dysfunction: a marker of atherosclerotic risk. Arterioscler Thromb Vasc Biol. 2003;23(2):168-175. doi:10.1161/01.atv.0000051384.43104.fc
  4. De Marchi S, Chiarioni G, Prior M, Arosio E. Young adults with coeliac disease may be at increased risk of early atherosclerosis. Aliment Pharmacol Ther. 2013;38(2):162-169. doi:10.1111/apt.12360
  5. Zhernakova A, Elbers CC, Ferwerda B, et al. Evolutionary and functional analysis of celiac risk loci reveals SH2B3 as a protective factor against bacterial infection. Am J Hum Genet. 2010;86(6):970-977. doi:10.1016/j.ajhg.2010.05.004”

The quality and depth of our discussion have been significantly augmented due to the reviewer's esteemed advice. The reviewer's continued guidance and support are truly crucial in enhancing the quality of this manuscript.

Reviewer 3:

I have only a minor comment: Blank page no. 9 should be removed before the publication.

The reviewer's meticulous attention to detail is greatly appreciated. The extraneous blank page, as kindly noted by the reviewer, has been removed from the manuscript. The adherence to such high standards of precision, as demonstrated by the reviewer, has contributed significantly to the readiness of our paper for publication. Thank you for your invaluable guidance and support

Reviewer 2 Report

Its a well written review.

There are minor corrections as listed below:

1. Different fonts have ben used in abstract.

2. Line 68-69: CD pathophysiology is not discussed in detail. Since its a review focusing on CD, emphasis should be given on HLA-DQ2/ DQ8

3. Line 95-96: space issue needs to rectified

4. In many places 'Celiac disease' is used instead of CD and vice versa. Please keep it consistent (with CD). Same applies to CI/ confidence interval

5. Molecular mechanism linking CD and CVD need to be explained in detail. 

Author Response

(The authors gave the same response as above.)

Reviewer 3 Report

It is an interesting review study on Celiac Disease and the Risk of Cardiovascular Diseases. I have only a minor comment: Black page no. 9 should be removed before the publication. 

Author Response

(The authors gave the same response as above.)
